# Incidence of switching to second-line antiretroviral therapy and its predictors among children on antiretroviral therapy at general hospitals, Northern Ethiopia: A survival analysis

**Migbar Mekonnen Sibhat** [1] *, **Tewodros Nigussie Mulugeta**[1], **Dawit W/tsadik Aklilu**[2]

**1** College of Health Sciences and Medicine, Dilla University, Dilla, Ethiopia, **2** College of Health Science, Debre Berhan University, Debre Berhan, Ethiopia

* bayayibignabez@gmail.com

**Data Availability Statement:** All relevant data are available within the paper and its Supporting Information files.

## Abstract

### Background

With expanding access to pediatric antiretroviral therapy, several patients in the developing world were switched to the second-line regimen, and some require third-line medications. A delay in a second-line switch is associated with an increased risk of mortality and other undesired therapeutic outcomes, drives up program costs, and challenges the pediatric antiretroviral therapy service. Nevertheless, there remain limited and often conflicting estimates on second-line antiretroviral therapy use during childhood, especially in resource-limited settings like Ethiopia. Thus, this study intended to determine the incidence and predictors of switching to second-line antiretroviral therapy among children.

### Methods

A retrospective cohort study was conducted by reviewing records of 424 randomly selected children on first-line antiretroviral therapy from January 2014 to December 2018 at public hospitals in the Central and Southern Zones of Tigray, Northern Ethiopia. Data were collected using extraction tool; entered into Epi-data; cleaned, and analyzed by STATA version-14. Kaplan-Meier curve, log-rank test, and life table were used for data description and adjusted hazard ratios and p-value for analysis by Cox proportional hazard regression. Variables at a P-value of ≤0.20 in the bi-variable analysis were taken to multivariable analysis. Finally, statistical significance was declared at a P-value of ≤0.05.

### Results and conclusion

Analysis was conducted on 424 charts with a total person-time observation of 11686.1 child-months and an incidence switch rate of 5.6 (95%CI: 4.36–7.09) per 1000 child-month-observations. Being orphan [AHR = 2.36; 95%CI: 1.10–5.07], suboptimal adherence [AHR = 2.10; 95% CI: 1.12–3.92], drug toxicity [AHR = 7.05; 95% CI: 3.61–13.75], advanced latest

**Funding:** This research received no specific grant from any funding agency in the public, commercial, or not-for-profit sectors. However, Dilla University covered the data collection cost. The funder had no role in study design, data collection, analysis, preparation of the manuscript, and decision to publish.

**Competing interests:** The authors have declared that no competing interests exist.

**Abbreviations:** ABC, abacavir; AIDS, acquired immune deficiency syndrome; ART, antiretroviral therapy; BSc, bachelor of science; MSc, master of science; EFZ, Efavirenz; HFA, height for age; HIV, human immunodeficiency virus; LMIC, low and middle-income countries; LPV/r, lopinavir/ritona; NNRTIs, nonnucleoside reverse transcriptase inhibitors; NRTIs, nucleoside reverse transcriptase inhibitors; NVP, nevirapine; OI, opportunistic infections; PI, protease inhibitors; SAM, severe acute malnutrition; TB, tuberculosis; URTI, upper respiratory tract infection; UTI, urinary tract infection; VL, viral load; WFA, weight for age; WHO, world health organization.

clinical stage [AHR = 2.75; 95%CI: 1.05–7.15], and tuberculosis co-infection at baseline [AHR = 3.08; 95%CI: 1.26–7.51] were significantly associated with switch to second-line antiretroviral therapy regimen. Moreover, a long duration of follow-up [AHR = 0.75; 95% CI: 0.71–0.81] was associated with decreased risk of switching. Hence, it is better to prioritize strengthening the focused evaluation of tuberculosis co-infection and treatment failure with continuous adherence monitoring. Further research is also needed to evaluate the effect of drug resistance.

## Introduction

The short-term effectiveness of Anti-Retroviral Therapy (ART) among children is undisputed [1–3]. Several patients in the developing world were switched to second-line therapy, and some require third-line medications [4–6]. Highly Active Antiretroviral Therapy (HAART) refers to drugs that serve to increase the life expectancy of children infected with the Human Immunodeficiency Virus (HIV) [7, 8].

There remain limited and often conflicting estimates on the use of second-line ART in children globally, ranging from 2–35% switching at 2–5 years after ART initiation [9–11]. Nonetheless, concerns have been raised that patients may be experiencing long periods of virologic failure [12, 13]. Many individuals who failed for first-line ART in sub-Saharan Africa never initiate second-line ART or do so after a significant delay [11, 14]. Delays in shifting to second-line ART are frequently noted among HIV-infected children in Low and Middle-Income Countries (LMIC) [15, 16], even after first-line treatment failure. Therefore, determining the second-line switching time and its predictors would be paramount to set an effective plan of care.

The recognized barriers to postponing second-line switch include but are not limited to inadequate access to standard treatment monitoring strategies, non-palatability of formulations for children, and limited access to second-line treatment options. Additionally, residing in low-income countries, dependency on caregivers to take medications, caregivers' literacy, and pediatric-care providers' intention to preserve restricted future treatment options [17, 18].

Accordingly, children could remain on failing regimens for longer than adults in the same position, putting them at risk of further accumulation of resistance mutations to backbone regimens while their health declines [17]. A delay in switching increases mortality [19] and the risk of developing opportunistic infections [14, 20]. Prolonged treatment with a failed regimen could result in a 46% raised chance of failure to second-line therapy [20, 21], increased drug toxicity [22], and increased drug resistance [23]. Consequently, this may end up with exhaustion to available treatment [11], drive up program costs [6, 24, 25], and challenges the ART service delivery in the pediatric age group [22]. On the other hand, failure to initiate the appropriate ART regimen may lead to neurodevelopmental retardation, malnutrition, and stunting [26, 27]. Thus, determining switching time is essential to prevent delay and minimize such unnecessary costs.

Currently, guidelines and panels recommend performing viral load testing at least every three months, intensive follow-up, simplifying the drug regimen, and caregiver education and engagement in child ART care to increase the sustainability and effectiveness of the pediatric ART services [7, 12].

Ethiopia is one of the high HIV-burden countries and does not have appropriate ART drug formulations for children beyond the 2nd line [12]. A solitary study in Ethiopia at Black Lion

hospital regarding this area of study, which determined only the incidence of the switch to second-line ART, reported that among those children who failed to respond to the first-line regimen, 14.4% were switched to second-line ART with a mean delay of 24 months [20].

Despite this significant delay in switching and scarcity of data related to this topic, we could not find studies conducted to evaluate factors that predict switching to second-line ART regimens among children throughout the country. Hence, this study was designed to assess second-line ART switch and its predictors among children who had been taking first-line ART.

## Methods and materials

### Study setting and population

An institution-based retrospective cohort study was employed by reviewing recordings of HIV/AIDS infected children who started to receive ART at the public hospitals of Central and Southern Zones of Tigray, Northern Ethiopia. Five general public hospitals provide ART care services in the study area (Mekelle General Hospital, Quiha General Hospital, Alamata General Hospital, Maichew General Hospital, and Korem General Hospital), which are eligible for selection. The study population comprised all recordings of HIV/AIDS infected children before their 15th birthday who started taking first-line ART from January 2014 to December 2018 at selected public general hospitals in Northern Ethiopia. Child records with incomplete documentation and no follow-up data after ART initiation were excluded since we could not obtain sufficient information to declare the event of interest.

### Sample size determination and sampling procedure

The sample size was determined using the Cox model according to the Cox proportional hazard model assumptions by using STATA version 14. The nevirapine-based regimen was used as a predictor variable from the previous study [28], which yields the maximum sample size. The following parameters were used: a 95% confidence interval, 80% power, hazard ratio (HR) = 2.47, probability of switch = 0.0641, and design effect of 1.5. After adding 5% contingency, the required sample size was 424. Thus, 424 child records needed to be incorporated for review to conduct this study. Afterward, the hospitals were selected using a cluster sampling technique considering each hospital (five hospitals) as a cluster (arranged as cluster 1 to cluster 5). Among those clusters (the five hospitals), the lottery method was used to select the three hospitals (Mekelle general hospital, Alamata general hospital, and Lemlem Karl hospital). Finally, five years of ART data (from January 2014 to December 2018) were reviewed from recordings of all children receiving first-line ART and followed-up at the three selected hospitals (Fig 1).

### Operational definitions and measurements

Switching to second-line ART (event) was defined as: (i) the commencement of ≥2 new drugs including a class-switch from Protease Inhibitors (PI) to Non-Nucleoside Reuptake Inhibitors (NNRTI) or vice versa irrespective of the cause, (ii) addition of new drug class, or (iii) change of ≥1 Nucleoside Reuptake Inhibitors (NRTIs) or change from Ritonavir (RTV) to Lopinavir (LPV/r) with reason documented as treatment failure [8, 12, 29]. **Censored** were those who did not switch to the second-line regimen during follow-up, including lost, transferred out, died, exceed 15th birthday during follow-up, and on first-line at the end of follow-up. The time scale was measured in months from ART start until the earliest switch or censor. A patient was considered a defaulter if there is no follow-up visit for ≥3 months. **Adherence** was measured based on the 2017 national ART score cut-offs [8, 12] using patient self-reporting by asking the child's caregiver:

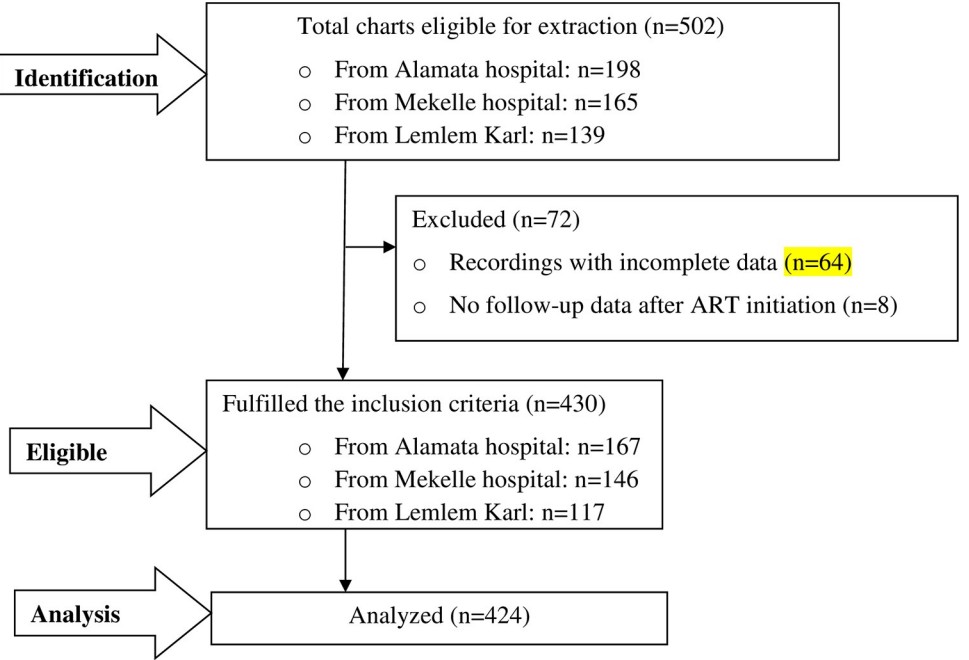

**Fig 1. Flow diagram showing sampling and participant selection procedure.**

☞ Good:—if the patient missed ≤ 2 doses in daily dose or ≤ 3 doses in bid regimens.

☞ Fair:—if missed 3–5 doses in a daily dose or 3–9 doses in twice a day preparations

☞ Poor:—if missed ≥ 6 doses in daily dose or ≥ 9 tabs in double dose regimens

## Data collection instruments and procedure

A data extraction checklist was used to collect the data after developed from the national HIV treatment guideline [12], ART monitoring chart, and related articles. The tool comprised socio-demographic, clinical and laboratory-related, treatment-related, and other factors. The lists of participants were taken from the ART data clerk, and unique ART numbers were used to find charts from the hospital card room. Four data collectors and three supervisors were recruited, and the data collection was accomplished from April 1–26 /2019.

## Data processing, analysis, interpretation, and presentation

Data were coded, cleaned, and entered into Epi-data. Afterward, analysis was done using STATA V-14. Then, the data was declared as survival-time data and described using frequency tables, percent, and median. Kaplan Meier's curve was considered to estimate median survival time during the follow-up period. Life-table was used to estimate the cumulative switch probabilities at different time intervals (S2 File, S1a and S1b Table). The incidence rate with a 95% confidence interval and cumulative incidence of switching was calculated.

Cox proportional hazard regression model was used to analyze the data. Those variables having $P \leq 0.20$ in the bi-variable analysis were included in the multivariable analysis. Proportional hazard model assumptions were checked using the Schoenfield residual test ($p = 0.26$). Schoenfield residual test outputs for the individual covariates were provided somewhere else (S3 Table in S2 File). Harrell's $C$ was also computed $(C = 0.9935)$, which indicates that this

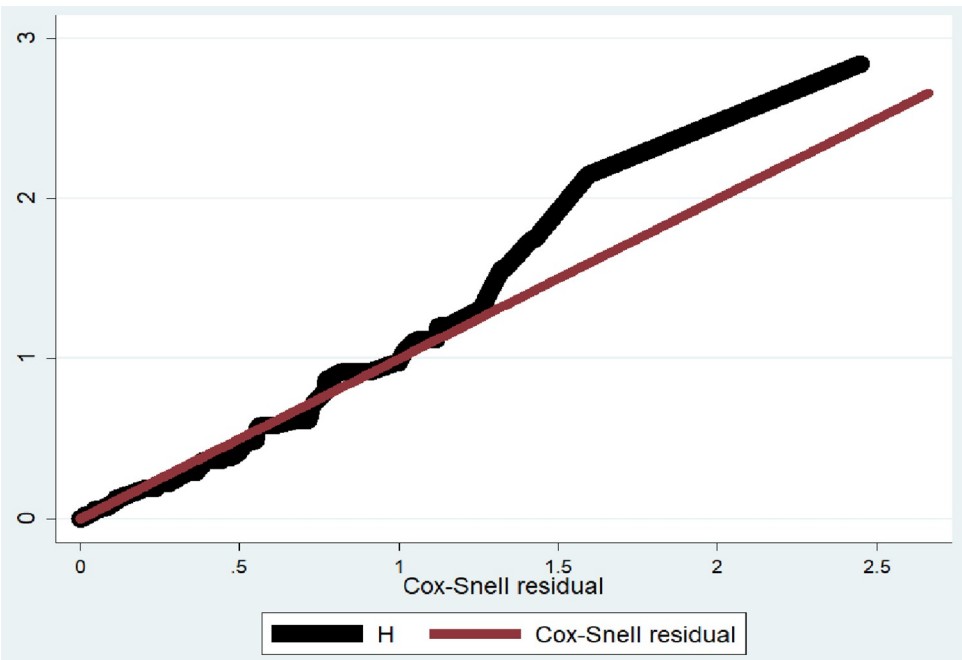

**Fig 2. Cox-Snell residual cumulative hazard graph for switching to second-line antiretroviral therapy and its predictors among HIV/AIDS infected children in public general hospitals, Northern Ethiopia, 2019/20 (n = 424).**

study can correctly order survival times for pairs of children 99.3% of the time on the basis of observations of fitted variables in the model. Furthermore, Cox regression model for its fitness to the data was checked using Cox-Snell residuals; in which the hazard function follows the 45 degree line very closely except for large values of time (Fig 2). Generally we could conclude that the final model fits the data successfully. In the multivariable analysis, statistical significance was declared at a p-value of ≤0.05, and the strength of association was reported using an adjusted hazard ratio with a 95% confidence interval. Finally, results were presented using texts, tables, and graphs.

## Data quality assurance

Data collectors and supervisors received one-day training. A pretest was conducted on 20 (5%) randomly selected charts at Ayder Comprehensive Specialized Hospital in order to check clarity and consistency of the tool so that unclear and confusing items were modified as needed. The collected data were audited daily by the principal investigator and supervisor. Whenever there is incompleteness and uncertainty in recording, the filled information was cross-checked with source data soon.

**Ethical approval and consent to participate.** Ethical approval was obtained (ERC 1272/ 2019) from the institutional review board (IRB) of Mekelle University, college of health sciences. The IRB waived such that the research could be done by record review without contacting patients since the study was conducted through a review of medical records. Permission letters were obtained from each hospital administration and respective hospital ART coordinators. All information was kept confidential, and no individual identifiers were collected.

## Results

An overall 502 children started the first-line ART regimen in the specified period. Of those, four-hundred twenty-four (n = 424) children aged less than 15 years who fulfilled the eligibility criteria were incorporated for analysis.

### Socio-demographic characteristics

Children were followed for a minimum of six months and a maximum of 60 months, with a median follow-up of 24.4 (IQR = 36.2) months. The study finding showed that two hundred eleven (49.8%) children were males, of which 16.6% fulfilled the definition of second-line switch (switch criteria), while this was true among 14.4% of female participants. The median age of children at ART initiation was nine (IQR = 9) years. A significant proportion (23.3%) of children aged 5–10 years met at least one of the switch definition/criteria, followed by >10 years of age (14.6%). The study result also revealed that 154 (36.3%) children were orphaned who lost either one or both parents, of which 34 (22.1%) were eligible for second-line ART regimens (Table 1).

### Clinical and laboratory-related characteristics

The study finding notified that 50% of the participants had CD4 count of less than 475cells/mm3 at the initiation. Again, among 111 (26.2%) children who started ART with advanced WHO clinical stage, 25 (22.5%) were identified to be switched to second-line ART regimens. Whereas, 20.4% of children who started with a CD4 count of less than 200cells/mm$^3$ fulfilled the switch criteria. Meanwhile, 201 (47.4%) children had no access to a viral load investigation (Table 2).

Considerably, study participants acquired opportunistic infections at baseline and after ART initiation were 38.9% and 25.2% respectively. Moreover, 20.6% and 27.1% of children having OI at baseline and after initiation respectively had switched to second-line ART regimens (Table 3).

### Treatment-related and other factors

Three hundred forty-three (80.9%) participants started with NNRTIs-based ART regimen. Two-hundred (47.2%) children started with NVP-based regimen followed by 46 (10.8%, 46/

**Table 1. Distribution of socio-demographic characteristics to assess second-line switching among of HIV/AIDS infected children on first-line ART in public general hospitals, Northern Ethiopia, 2019/20, (n = 424).**

| Independent variables | Categories | ART Outcome | | |
|---|---|---|---|---|
| | | Switched | Not switched | Total |
| | | Count (%) | Count (%) | Count (%) |
| Child age at ART initiation | <5 years | 6 (5.5) | 104 (94.5) | 110 (25.9) |
| | 5–10 years | 35 (23.3) | 115 (76.7) | 150 (35.4) |
| | >10 years | 24 (14.6) | 140 (85.4) | 164 (38.7) |
| Sex of the child | Female | 30 (14.1) | 183 (85.9) | 213 (50.2) |
| | Male | 35 (16.6) | 176 (83.4) | 211 (49.8) |
| Parent status | Both alive | 31 (11.5) | 239 (88.5) | 270 (63.7) |
| | Either Died | 19 (19.4) | 79 (80.6) | 98 (23.1) |
| | Both Died | 15 (26.8) | 41 (73.2) | 56 (13.2) |

**Abbreviation:** ART, antiretroviral therapy

**Table 2. Clinical and laboratory-related characteristics to assess second-line switching among of HIV/AIDS infected children on first-line ART in public general hospitals, Northern Ethiopia, 2019/20, (n = 424).**

| Independent variables | Category | ART Outcome | | Total (%) |
|---|---|---|---|---|
| | | Switched | Not switched | |
| | | Count (%) | Count (%) | |
| WFA at baseline | < 3$^{rd}$ | 39 (15.7) | 209 (84.3) | 248 (58.5) |
| | 3rd - 97$^{th}$ | 26 (15.2) | 145 (84.8) | 171 (40.3) |
| | > 97$^{th}$ | 0 | 5 (100) | 5 (1.2) |
| HFA at baseline | < 3rd | 54 (15.7) | 289 (84.3) | 343 (80.9) |
| | 3rd - 97th | 6 (11.1) | 48 (88.9) | 54 (12.7) |
| | > 97th | 5 (18.5) | 22 (81.5) | 27 (6.4) |
| WHO stage at ART start | Early | 40 (12.8) | 273 (87.2) | 313 (73.8) |
| | Advanced | 25 (22.5) | 86 (77.5) | 111 (26.2) |
| WHO stage at last visit | Early | 56 (14.2) | 338 (85.8) | 394 (92.9) |
| | Advanced | 9 (30) | 21 (70) | 30 (7.1) |
| CD4 count at baseline | ≤200 | 10 (20.4) | 39 (79.6) | 49 (11.6) |
| | >200 | 42 (16.3) | 216 (83.7) | 258 (60.8) |
| | Unknown | 13 (11.1) | 104 (88.9) | 117 (27.6) |
| Latest CD4 count | ≤200 | 15 (53.6) | 13 (46.4) | 28 (6.6) |
| | >200 | 26 (12.9) | 175 (87.1) | 201 (47.4) |
| | Unknown | 24 (12.3) | 171 (87.7) | 195 (46) |
| Access to viral load | No | 14 (7) | 187 (93) | 201 (47.4) |
| | Yes | 51 (22.9) | 172 (77.1) | 223 (52.6) |
| VL at initiation | <1000c/ml | 6 (13.6) | 38 (86.4) | 44 (10.4) |
| | ≥1000c/ml | 7 (35) | 13 (65) | 20 (4.7) |
| | Unknown | 52 (14.4) | 308 (85.6) | 360 (84.9) |
| Latest VL | <1000c/ml | 23 (14.2) | 139 (85.8) | 162 (38.2) |
| | ≥1000c/ml | 28 (47.5) | 31 (52.5) | 59 (13.9) |
| | Unknown | 14 (6.9) | 189 (93.1) | 203 (47.9) |

**Abbreviations:** ART, antiretroviral therapy; WFA, weight for age; HFA, height for age; WHO, world health organization; VL, viral load; c/ml, copies per milliliter

424) with ABC-based and 35 (8.3%, 35/424) with boosted PI-based regimens. Sixty-four (15.1%) children had previous ART exposure. On the other hand, 31 (7.3%) children did not take any OI prophylaxis. One hundred twenty (28.3%) children had not disclosed their serostatus on ART start; 117 (27.6%) had suboptimal adherence as well as 133 (31.5%) and 101 (23.8%) developed adverse effects and substituted their initial first-line regimen during follow-up respectively (Table 4).

Moreover, 65 (15.33%) children satisfied at least one switch criterion/definition, of which only 31 (47.7%) switched to second-line regimens. The rest 359 (84.67%) children were censored, with seven (1.7%) died during follow-up, 61 (14.4%) transferred out to other facilities, and 35 (8.2%) lost during the follow-up period. Additionally, two-thirds (64.6) of the children switched to second-line regimen were attributable to first-line treatment failure.

## Comparison of survival status using Kaplan-Meier curve

The Kaplan Meier switch curve increased stepwise as the follow-up time increased, and it crosses the survival function at a survival probability of 0.5 just before 60 months (Fig 3). This implies, as the follow-up time increases, the probability of children to switch to second-line ART regimen increased as well. The Kaplan-Meier survival probability plots for each

**Table 3. Distribution of opportunistic infections to assess second-line switching among of HIV/AIDS infected children on first-line ART in public general hospitals, Northern Ethiopia, 2019/20, (n = 424).**

| Covariates | Category | | ART Outcome | | |
| --- | --- | --- | --- | --- | --- |
| | | | Switched | Not switched | Total (%) |
| | | | Count (%) | Count (%) | |
| Opportunistic infections at baseline | No | | 31 (12) | 228 (88) | 259 (61.1) |
| | Yes | Anemia | 1 (6.7) | 14 (93.3) | 15 (3.5) |
| | | Diarrhea | 13 (20.3) | 51 (79.7) | 64 (15.1) |
| | | SAM | 6 (17.1) | 29 (82.9) | 35 (8.3) |
| | | TB | 14 (27.5) | 37 (72.5) | 51 (12) |
| | | Pneumonia | 19 (25) | 57 (75) | 76 (17.9) |
| | | URTI | 6 (24) | 19 (76) | 25 (5.9) |
| | | UTI | 3 (75) | 1 (25) | 4 (0.9) |
| | | Candidiasis | 1 (33.3) | 2 (66.7) | 3 (0.7) |
| | | Oral thrush | 2 (20) | 8 (80) | 10 (2.4) |
| | | Meningitis | 1 (20) | 4 (80) | 5 (1.2) |
| | | Total | 34 (20.6) | 131 (79.4) | 165 (38.9) |
| Opportunistic infections after ART initiation | No | | 36 (11.4) | 281 (88.6) | 317 (74.8) |
| | Yes | Anemia | 9 (34.6) | 17 (65.4) | 26 (6.1) |
| | | Diarrhea | 6 (21.4) | 22 (78.6) | 28 (6.6) |
| | | SAM | 6 (22.2) | 21 (77.8) | 27 (6.4) |
| | | TB | 10 (31.2) | 22 (68.8) | 32 (7.5) |
| | | Pneumonia | 13 (31.7) | 28 (68.3) | 41 (9.7) |
| | | URTI | 6 (31.6) | 13 (68.4) | 19 (4.5) |
| | | UTI | 0 | 4 (100) | 4 (0.9) |
| | | Candidiasis | 2 (22.2) | 7 (77.8) | 9 (2.1) |
| | | Oral thrush | 1 (50) | 1 (50) | 2 (0.5) |
| | | Meningitis | 3 (42.9) | 4 (57.1) | 7 (1.7) |
| | | Total | 29 (27.1) | 78 (72.9) | 107 (25.2) |

**Abbreviations:** ART, antiretroviral therapy; SAM, severe acute malnutrition; TB, tuberculosis; URTI, upper respiratory tract infection; UTI, urinary tract infection

independent variables was presented somewhere else (S1 and S2 Figs in S1 File). The log-rank test for equality of survivor functions between groups was provided somewhere (S2 Table in S2 File).

## Survival function and an incidence rate of second-line ART switch

The total child-month observation was 11686.1 child-months with the incidence switch rate of 5.6 (95% CI 4.36–7.09) per 1000 child-months of observation. The median survival time was found to be 58.7 months. The cumulative probabilities of switch at 12, 24, 36, 48 and 60 months were 0.053, 0.08, 0.13, 0.24 and 0.52 respectively (S1 Table in S2 File).

## Predictors of switching to second-line ART

Some variables such as WFA, HFA, recent CD4 count, baseline viral load, drug substitution, and OI prophylaxis were left out of the final model since they have less than 20% predicted events per cell. Afterward, in the final Cox proportional hazard model, being orphaned, suboptimal ART adherence, drug toxicity, advanced recent WHO stage, having TB co-infection at

**Table 4. Distribution of treatment-related and other factors to assess second-line switching among of HIV/AIDS infected children on first-line ART in public general hospitals, Northern Ethiopia, 2019/20, (n = 424).**

| Independent variables | Category | | What was the last outcome? | | |
|---|---|---|---|---|---|
| | | | Switched | Not switched | Total (%) |
| | | | Count (%) | Count (%) | |
| Previous ART exposure | No | | 61 (16.9) | 299 (83.1) | 360 (84.9) |
| | Yes | | 4 (6.2) | 60 (93.8) | 64 (15.1) |
| OI prophylaxis | No | | 1(3.2) | 30 (96.8) | 31 (7.3) |
| | Yes | | 64 (16.3) | 329 (83.7) | 393 (92.7) |
| Disclosure status | No | | 18 (15) | 102 (85) | 120 (28.3) |
| | Yes | | 47 (15.5) | 257 (84.5) | 304 (71.7) |
| Adherence to ART | Sub-optimal | Poor | 16 (25.8) | 46 (74.2) | 62 (14.6) |
| | | Fair | 12 (21.8) | 43 (78.2) | 55(13) |
| | Optimal | Good | 37 (12.1) | 270 (87.9) | 307 (72.4) |
| Baseline ART regimen | NVP-based | | 37 (18.5) | 163 (81.5) | 200 (47.2) |
| | ABC-based | | 5 (10.9) | 41(89.1) | 46 (10.8) |
| | EFV-based | | 19 (13.3) | 124 (86.7) | 143 (33.7) |
| | LPV/r-based | | 4 (11.4) | 31(88.6) | 35 (8.3) |
| ART drug Toxicity | No | | 13 (4.5) | 278 (95.5) | 291 (68.6) |
| | Yes | | 52 (39.1) | 81(60.9) | 133 (31.4) |
| ART drug substitution | No | | 0 | 323 (100) | 323 (76.2) |
| | Yes | PI to NNRTI | 44 (100) | 0 | 44 (10.4) |
| | | <2 NRTI | 11 (57.9) | 8 (42.1) | 19 (4.5) |
| | | New class added | 10 (100) | 0 | 10 (2.3) |
| | | within NNRTI | 0 | 28 (100) | 28 (6.6) |
| Treatment failure | Yes | | 42 (43.8) | 54 (56.2) | 96 (22.6) |
| | No | | 23 (7) | 305 (93) | 328 (77.4) |

**Abbreviations:** ART, antiretroviral therapy; OI, opportunistic infections; PI, protease inhibitors; NNRTIs, nonnucleoside reverse transcriptase inhibitors; NRTIs, nucleoside reverse transcriptase inhibitors; NVP, nevirapine; ABC, abacavir; EFZ, Efavirenz; LPV/r, lopinavir/ritonavir

baseline, and duration of follow-up were found to be independent predictors of switching to a second-line ART regimen (Table 5).

## Discussion

This study aimed to determine the incidence and predictors of switching to a second-line ART regimen among HIV/AIDS infected children. The median survival time was 58.7 months, with an overall incidence switching rate of 5.6 (95% CI 4.36–7.09) per 1000 child-month-observations. Being orphan, suboptimal ART adherence, drug toxicity, advanced latest WHO stage, baseline TB infection, and duration of follow-up were found to be independent predictors of the second-line switch.

The overall cumulative incidence of switching at five years in this study was 52% (95% CI 39.61–66.34). This was higher than reports in previous studies; 31.6% in Asia-Pacific and African countries [30] and 21% in Europe and Thailand [28]. The possible explanation could be the advancements in diagnostic and therapeutic measures. These include more frequent visits, increased access to viral load, and availability of more potent drugs nowadays than in the past [12].

Having tuberculosis at ART initiation was significantly associated with switching to second-line ART drugs [AHR = 3.08; 95%CI: 1.26–7.51]. Despite published pediatric studies

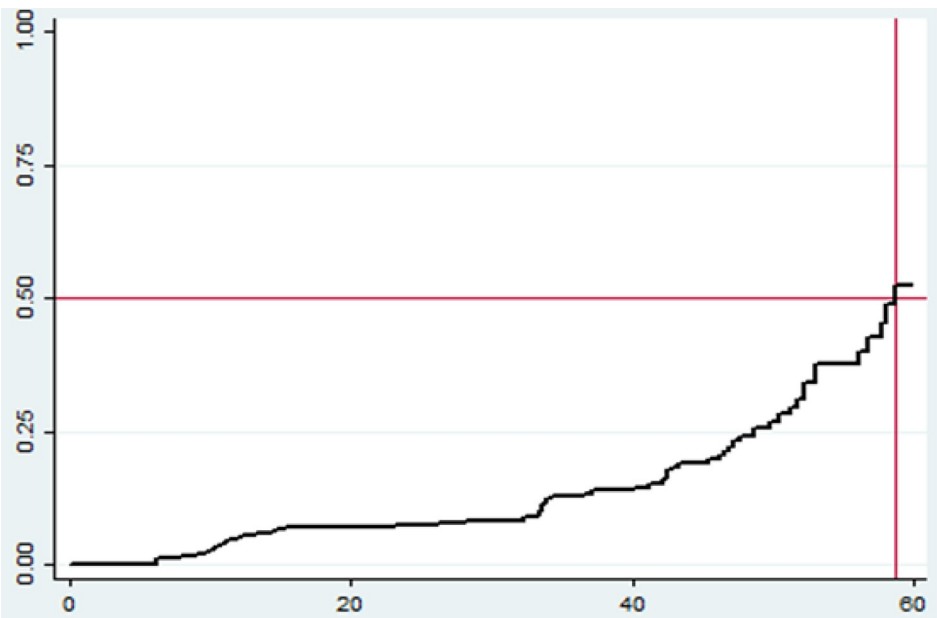

**Fig 3. Overall Kaplan Meier failure estimate of HIV/AIDS infected children in public general hospitals, Northern Ethiopia, 2020 (n = 424).** The Y-axis represents the probability of a second-line ART switch whereas the X-axis indicates the follow-up time in months. The red vertical and horizontal lines were reference lines added to ease graph interpretation (median time to switch estimation).

never reported tuberculosis co-infection as an independent predictor of a second-line switch, it was documented as the primary cause of first-line treatment failure [31, 32]. This could be rationalized as tuberculosis infection facilitates viral replication by activating the immune system, which leads to viral load increment. Again, this will accelerate rapid HIV/AIDS disease progression, which cross-reacts with the ART drug action [12]. This leads to increased replication of drug-resistant mutations and thus a higher chance of switching.

The other factor that showed a significant association was ART drug toxicity. Children who developed adverse effects were seven times at higher hazard of a switch than those with no drug toxicity during follow-up [AHR = 7.05; 95% CI: 3.61–13.75]. This finding was supported by a previous study conducted in West African countries [26]. This might be because intolerance to ART drugs is a barrier to adherence leading to treatment discontinuation, risking viral rebound, and drug resistance [7, 12]. The major cause of drug discontinuation in the first 3–6 months after ART initiation is drug toxicity.

The study finding also revealed that being an orphan child was significantly associated with second-line switch. The findings of a recent pediatric study conducted in Cameroon supported this association [33]. Children are dependent on their parents and require continual support due to their age and developmental stages. Parents play an undeniable role in providing effective pediatric ART service by encouraging children to take ART drugs timely, as prescribed, assist in cases of sub-optimal adherence, attend follow-up based on appointments, and avoid the sense of loneliness. On the contrary, orphaned children will miss doses and follow-up visits, have difficulty with adherence, and may not fully understand healthcare instructions. Besides, poor palatability of drugs, frequent adverse effects, limited formulations, and frequent dosing requirements [7, 12] may lead to poor adherence and exacerbate the level of negligence.

This study revealed that the advanced WHO stage at the last visit was linked with a higher hazard of a second-line switch than the early WHO stage [AHR = 2.75; 95%CI: 1.05–7.15].

**Table 5. Bivariate and multivariable Cox-proportional hazard regression analysis output for the predictors of second-line ART switch among children with HIV/AIDS in public general hospitals, Northern Ethiopia, 2019/20 (n = 424).**

| Covariates | Category | Last outcome | | Crude HR (95% CI) | Adjusted HR (95% CI) |
|---|---|---|---|---|---|
| | | Switched | Not switched | | |
| **Parent status** | Both alive | 31 | 239 | - | - |
| | Either died | 19 | 79 | 1.92 (1.08–3.41) | 2.13 (1.04–4.38)* |
| | Both died | 15 | 41 | 2.81 (1.52–5.22) | 2.36 (1.10–5.07)* |
| **OI at baseline** | Yes | 34 | 131 | 1.87 (1.15–3.05) | 1.15 (0.56–2.37) |
| | No | 31 | 228 | - | - |
| **OI after ART initiation** | Yes | 29 | 78 | 2.47 (1.51–4.04) | 1.77 (0.77–4.09) |
| | No | 36 | 281 | - | - |
| **Adherence to ART drugs** | Optimal | 37 | 270 | - | - |
| | Sub-optimal | 28 | 89 | 2.51 (1.53–4.11) | 2.10 (1.12–3.92)* |
| **Previous ART exposure** | Yes | 4 | 60 | - | - |
| | No | 61 | 299 | 1.90 (0.69–5.23) | 0.43 (0.14–1.31) |
| **ART drug toxicity** | Yes | 52 | 81 | 5.68 (3.08–10.49) | 7.05 (3.61–13.75)* |
| | No | 13 | 278 | - | - |
| **Baseline WHO stage** | Early-stage | 40 | 273 | - | - |
| | Advanced stage | 25 | 86 | 1.91 (1.16–3.16) | 1.00 (0.50–2.00) |
| **Latest WHO stage** | Early-stage | 56 | 338 | - | - |
| | Advanced stage | 9 | 21 | 4.20 (2.06–8.59) | 2.75 (1.05–7.15)* |
| **Baseline CD4** | Failed | 10 | 39 | 1.72 (0.86–3.43) | 0.83 (0.36–1.92) |
| | Normal | 42 | 216 | - | - |
| | Unknown | 10 | 39 | - | - |
| **Anemia after ART start** | No | 56 | 342 | - | - |
| | Yes | 9 | 17 | 4.51 (2.18–9.34) | 0.22 (0.05–1.02) |
| **Diarrhea after ART initiation** | No | 59 | 337 | - | - |
| | Yes | 6 | 22 | 1.95 (0.84–4.54) | 0.82 (0.26–2.62) |
| **Malnutrition at baseline** | No | 59 | 330 | - | - |
| | Yes | 6 | 29 | 1.81 (0.78–4.23) | 2.31 (0.78–6.89) |
| **Malnutrition after ART start** | No | 59 | 338 | - | - |
| | Yes | 6 | 21 | 2.45 (1.05–5.73) | 0.40 (0.11–1.53) |
| **TB at baseline** | No | 51 | 322 | - | - |
| | Yes | 14 | 37 | 4.03 (2.21–7.38) | 3.08 (1.26–7.51)* |
| **TB after ART initiation** | No | 55 | 337 | - | - |
| | Yes | 10 | 22 | 1.79 (0.91–3.55) | 2.23 (0.83–5.98) |
| **Follow up duration** | | 65 | 359 | 0.89 (0.88–0.91) | 0.75 (0.71–0.81)* |

Note:

*significant at 5% level of significance

**Abbreviations:** OI, opportunistic infection; ART, antiretroviral therapy; WHO, world health organization; CD4, HIV helper cell count; TB, tuberculosis

Earlier pediatric studies did not examine the WHO stage at the last follow-up visit. However, previous studies investigated the WHO stage at baseline and reported it as an independent predictor of switching, which was not found statistically significant in the current study. The possible justification for the discrepancy could be due to advancements in the management strategies and short follow-up visit recommendations in the recent ART guideline [12]. Consequently, it will enable early detection, control, and management of adverse events and opportunistic infections. It can also help in the treatment regimen selection, diagnostic and monitoring workups, and repeated follow-up visits.

The current study also showed that the hazard of switching among children started ART follow-up with sub-optimal adherence was doubled compared to their counterparts [AHR = 2.10; 95% CI: 1.12–3.92]. This report was in agreement with the previous study result [26]. The possible reason for this could be due to the role of a high level of sustained adherence to ART treatment outcomes. Optimal adherence is necessary to reduce the risk of ART drug resistance and decrease the chance of HIV transmission by suppressing viral replication and improving immunological and clinical outcomes. On the other hand, HIV/AIDS infected children and adolescents frequently come with poor adherence to ART drugs. Several reasons could be stated for this, including limited choice of pediatric ART formulations, poor palatability of some drug preparations, the requirement of multiple pills with frequent dosing, as well as occurrence of potential adverse effects, and drug interaction in pediatric regimens. Adherence could also be affected by the child's age and developmental stage. This is because children need support from others to take medication timely and may face difficulties swallowing tablets [7, 12].

A long duration of follow-up was also related to decreased risk of switching to second-line ART [AHR = 0.75; 95% CI: 0.71–0.81]. This finding contradicts the results of a West African study [26]. The possible justification could be the West African study considered children that experienced first-line ART failure and estimated switch to second-line among only those who failed for first-line ART drugs. The other possible rationale for this association might be children who had been on ART for a prolonged period will have improved adherence and adaptation. This is because the level of adherence in children increases with time, and the need for adaptation to daily ART drug intake in the early ART follow-up periods may hamper the ART response, leading to inadequate viral suppression and thus the emergence of resistant mutations [7]. This may also be associated with the increased chance of occurrence of different rapid effects such as IRIS within the early months of ART initiation.

Although this study reported pertinent findings by considering censored observations and time data for analysis, and used longer follow-up period to estimate incidence of switch, it has certain limitations. First, since the study involves review of patient recordings, the general limitation of retrospective studies should be reminded-of while applying the study findings. For instance, we could not analyze the effect of drug resistance on second line switching since there was no documented data on this issue. Nevertheless, it was indicated as a pertinent variable in existing literature on the area of ART and, thus, future researches are required to evaluate the effect of drug resistance. Second, excluding charts with missing data and charts absent during the data collection period might under or overestimate the study findings.

## Conclusion

The overall cumulative incidence of the second-line switch was higher than in previous studies. A remarkable delay in switching to second-line ART drugs was observed. Furthermore, children who had ART drug toxicity, TB co-infection at ART initiation, advanced WHO clinical stage after ART initiation, non-adherence to ART regimen, those who were orphaned, and on ART for a short period were at higher hazard of switching.

## Supporting information

**S1 File. Supporting figures.**
(DOCX)

**S2 File. Supporting tables.**
(DOCX)

**S1 Dataset.**
(DTA)

## Acknowledgments

The authors would like to appreciate data collectors, supervisors, hospital staff, and administrators for their unreserved efforts and commitment. Besides, the authors acknowledged Dilla University for covering the data collection cost.

## Author Contributions

**Conceptualization:** Migbar Mekonnen Sibhat, Dawit W/tsadik Aklilu.

**Data curation:** Migbar Mekonnen Sibhat, Tewodros Nigussie Mulugeta, Dawit W/tsadik Aklilu.

**Formal analysis:** Migbar Mekonnen Sibhat, Tewodros Nigussie Mulugeta.

**Funding acquisition:** Migbar Mekonnen Sibhat, Tewodros Nigussie Mulugeta.

**Investigation:** Migbar Mekonnen Sibhat.

**Methodology:** Migbar Mekonnen Sibhat, Dawit W/tsadik Aklilu.

**Project administration:** Migbar Mekonnen Sibhat.

**Resources:** Migbar Mekonnen Sibhat, Tewodros Nigussie Mulugeta, Dawit W/tsadik Aklilu.

**Software:** Migbar Mekonnen Sibhat.

**Supervision:** Migbar Mekonnen Sibhat, Tewodros Nigussie Mulugeta, Dawit W/tsadik Aklilu.

**Validation:** Migbar Mekonnen Sibhat.

**Visualization:** Migbar Mekonnen Sibhat, Tewodros Nigussie Mulugeta, Dawit W/tsadik Aklilu.

**Writing – original draft:** Migbar Mekonnen Sibhat.

**Writing – review & editing:** Migbar Mekonnen Sibhat, Tewodros Nigussie Mulugeta, Dawit W/tsadik Aklilu.

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
