## [Decision Letter · Decision Letter 0]

27 Mar 2023

PONE-D-22-19688Incidence of switching to second-line antiretroviral therapy and its predictors among children with HIV/AIDS at general hospitals, Northern Ethiopia : A survival analysisPLOS ONE

Dear Mr. Migbar Sibhat,

Thank you for submitting your manuscript to PLOS ONE. After careful consideration, we feel that it has merit but does not fully meet PLOS ONE’s publication criteria as it currently stands. Therefore, we invite you to submit a revised version of the manuscript that addresses the points raised during the review process.

We look forward to receiving your revised manuscript.

Kind regards,

Dorina Onoya

Academic Editor

PLOS ONE

Reviewers' comments:

Reviewer's Responses to Questions

Comments to the Author

1. Is the manuscript technically sound, and do the data support the conclusions?

Reviewer #1: Yes

Reviewer #2: No

Reviewer #3: Partly

2. Has the statistical analysis been performed appropriately and rigorously? 

Reviewer #1: Yes

Reviewer #2: No

Reviewer #3: Yes

3. Have the authors made all data underlying the findings in their manuscript fully available?

Reviewer #1: Yes

Reviewer #2: Yes

Reviewer #3: Yes

4. Is the manuscript presented in an intelligible fashion and written in standard English?

Reviewer #1: Yes

Reviewer #2: No

Reviewer #3: No

5. Review Comments to the Author

Reviewer #1: Retrospective cohort study was conducted by reviewing eligible child recordings at the selected public hospitals in the Central and Southern Zones of Tigray, Northern Ethiopia. The study determined the incidence and predictors of switching to second-line antiretroviral therapy among children. The results presented are interesting with significant public health implications. This is a well written manuscript, and I would recommend for acceptance. Questions/Comments as follows:

Introduction:

The introduction contains great information but long, and would benefit from being condensed and streamlined.

Methods /results

Results have been presented very thoroughly but I recommend the following;

-Table 5: should be re-done to also show child-months, switch rate per 100 child-months. The crude estimates should be excluded and rather focus on the adjusted estimates

-Figures Kaplan Meier curves should be plotted for all the significant factors in the cox proportional hazards model and not just the failure estimates. (stratified by for example WHO stage, being orphan, drug toxicity etc.)

Discussion and conclusion

-Discussion explores predictive factors very thoroughly

For the sentence “In this study, the median survival time was 58.7 months which is longer than the findings of previous studies; 35 months from a global pooled estimate by CIPHER, and 30 months in Europe and Thailand. The possible explanation could be the advancements in diagnostic and therapeutic measures. These include more frequent visits, increased access to viral load, and availability of more potent drugs nowadays than in the past.” The author in his introduction alluded to the fact there were no studies conducted to evaluate factors tat predict switching to second-line ART regimens among children throughout the country so how does he justifies his possible explanations?

Preferably remove actual results in the discussion and rather discuss key finding

Discussion does not cover limitations of the study, there are several limitations associated with observational studies and should be considered within these findings

Reviewer #2: Thanks Dr for inviting me to revise PONE-D-22-19688

The authors addressed important public health problem with sound methodology. I have the following comments and views on your manuscript.

There are studies which addresses similar title in Ethiopia. What makes your different and added a new knowledge

There are also systematic review and meta-analysis studies .Abstract, method part: sampling technique and P-value for declaration of Statically significance were not reported

Comments to the authors

The statistical analysis suffers from the Table 2 fallacy, whereby multiple inferences are made using a single model. Moreover, it is very weird (actually wrong) that for all you used categorical predictors is

1. WFA at baseline, and

2. HFA at baseline because it dependents on age and the classification also varied for children less than five years and above five years also

3. You also missed MUAC being either <11.5 .

4. Also CD4 count at baseline has its classification based on WHO definition

Therefore, this manuscript has both stastical fallacy and predictors weird

What is you reference to code VL at initiation as below or above 100

Where is your Model fitness test if you used cox regression unless there is no indication in graphically or statically?

Method part: You said about data collection periods April 1-26 /2019 and However I have assessed on Table 1: as you said it was in 2020GC it seems miss-behave on scientific environment and the data collection was even I have accepted it how it it is long duration influence the representativeness of your findings

Model you used seems to me inappropriate as you used , please it is fact than treatment switching can be caused by more than one factors and hazard could not be constant , there for the brain of the research is missed and it is more efficient that predictor should to be identified by Computing risk regression rather than cox regression , or distribution of treatment fail in causes weird distribution both intrinsic and proximal risk factors , there for it also appropriated by Exponential regression is more also properties , there for cumbersome reading more than is because failed data management

There for

1. miss-behave data collection periods

2. Miss data analysis model selection

3. Miss Nutritional classification whom all <15 years WFA, HFA, and MUAC classification

4. I couldn’t tolerated here is enough

Reviewer #3: GENERAL COMMENTS

• First, I would like to appreciate the authors’ intention towards identifying predictors of switching to second-line antiretroviral therapy, which is a devastating condition globally. It gives a clue about what to do in order to reduce treatment failure.

• But the manuscript is with a lot of grammatical and spelling errors and thus needs to be thoroughly revised by a native English speaker.

• Both in the abstract and the full text, abbreviations should be written entirely in their first appearance. Then after, it could be better to use the abbreviation throughout the whole text.

• The authors should be in line with the journal submission guidelines.

• Kindly find below my specific comments to help improve the quality of the manuscript:

Topic: Better to revise the topic as, Incidence of switching to second-line therapy and its predictors among children on antiretroviral therapy at general hospitals, Northern Ethiopia: A survival analysis.

ABSTRACT:

Methods:

• This section should incorporate all the main methods you used in conducting the research. So, you have to add the sample size, the population of the study, the study period, and the sampling procedure to recruit study participants.

• Page 2, lines 34-35: The authors stated that “…… log-rank test, and life table were used for data description….”; but there is no any life-table and log-rank value in the result section of the main document. So, better to revise this.

• How did you check the model adequacy for your data?

Results and conclusion:

• Be in line with the authors’ submission guideline.

• Page 2, Line 42: “…… advanced latest WHO stage…” use either ‘advanced WHO stage’ or ‘latest WHO stage’.

• Page 2, lines 47-48: “Further research is also needed to evaluate the effect of drug resistance.” At the abstract section, please recommend based on your main findings. The recommendation under the quotation should be included in the main document.

INTRODUCTION:

• Since the introduction lacks idea flows, this section requires grammatical editing and rearrangements of sentences and paragraphs.

• Page 3, lines 57 – 62: “Switching to second-line ART was defined as changing ≥2 new drugs, ……. with the reason of switch documented as treatment failure”. I recommend the authors remove this definition from the introduction and incorporate it into the operational definition of the method section.

• Page 4, lines 85 – 86: “Shreds of evidence also suggested that the costs of second-line ART drugs were more than double to triple compared with first-line regimen costs”.

o What is the implication of this sentence in accordance with your objective? I recommend the author remove it.

• It is expected more from the author to show what other literature has reported regarding the incidence of switching to a second-line regimen.

• Page 4, lines 91 – 92: “Ethiopia is one of the high HIV-burden countries and does not have appropriate ART drug formulations for children beyond the 2nd line”

o How could the lack of ART drug formulations beyond the 2nd line be related to a delay in switching to 2nd-line ART?

• It is not recommended to use unusual words in research writing, such as Albeit.

METHODS AND MATERIALS

• You missed a sub-heading “study setting”.

• Be consistent in using phrases like “HIV/AIDS-infected children” vs “HIV/AIDS patient” throughout the document.

• When was the data collected? Why didn't you incorporate children who started ART after 2018?

• What type of cohort that the authors used? Was it open or closed cohort?

Sample size determination and sampling procedure:

• What does a design effect of 1.5 mean? Is there a fractional design effect?

• Was there any ground or reason to select three from five hospitals? Why not two of four?

• Page 5, lines 125 – 126: remove the description “The detail of the sampling and participant selection procedure was presented in the flowchart below”. Only (Figure 1) could be enough.

• Figure 1: The sum of the bracketed numbers (66 and 8) is 74, not 72.

o Why did the authors exclude those eight patients who didn't have follow-up data after ART initiation? Why didn’t you consider these patients as a censored?

o If 430 patients fulfilled the inclusion criteria, why have you only analyzed 424 patients? Why didn’t you survey all of them?

Operational definitions:

• Censored (Lines 131-134): Is there a distinction between 'those who did not switch to the second-line regimen during follow-up' and 'those who were still on first-line at the end of follow-up'?

o Also, the description “Children were at risk of switching from ART start until the earliest switch or censor” couldn’t be an operational definition. So, please remove it.

• Adherence (Lines 135-137): when did you say the patient’s level of adherence was either Good, Fair, or Poor? This should be revised.

Data processing and analysis:

• Clearly show the statistical model you used?

• Why did you compute Harrell’s C statistics? You have to clearly put in the document its role.

• Line 149-150: “Life-table was used to estimate the ……”; but there is no any life table in the document.

• Lines 152 – 153: “Those variables having P≤0.20 in the bi-variable analysis …” Vs “variables ≤0.20 in the univariate analysis (in the abstract section)”: make it consistent.

• Lines 153 – 154: “Proportional hazard model assumptions were checked using the Schoenfield residual test (p=0.26)”.

o Proportional hazard assumption is conducted for each exposure variable; that means you couldn’t put a common p-value for different variable.

o So, a single Schoenfield residual value of 0.26 should be of the overall fitness, not proportional hazard assumption for variables.

Data quality assurance:

• Where did pretest was conducted? Why?

• Lines 163-164: Since it is already stated in the eligibility criteria above, it is better to remove the description “Individual records with incomplete data were excluded from the analysis”.

RESULTS:

• Over all, the result section is full of grammatical errors and needs a thorough revision.

• Medians should be described along with their interquartile range (IQR).

• Page 7, Lines 178 – 180: the authors reported that “two hundred eleven (49.8%) children were males, of which 16.6% fulfilled the switch criteria”. What exactly does it mean by "fulfilled the switch criteria"?

• Line 189 - 191: “…. had less than 475cells/mm3” what type of cells? Please specify clearly in the document.

o “ …. advanced WHO clinical stage 25 (22.5%) were”, what to mean?

• Figure 2: why you prefer failure cure other than survival curve?

o Lines 225 – 226: “...... and it crosses the survival function at a survival probability of 0.5”; so, what it indicates?

• You repeatedly wrote the sentence “Variables with a p-value of ≤0.20 in the univariate analysis were transferred to the multivariable analysis after checking the model fitness and proportional assumptions.” So, please remove it from here.

• Table 5: I recommend to the authors to remove the columns with P-values because CI are enough to show that the variable was significantly associated.

o The authors should interpret and write in text the common and significantly associated variables.

• If the authors computed a Cox-Snell residual test (as stated in the method section), where is the graph?

DISCUSSION:

• Page 13, lines 248 – 253: since this paragraph has already been stated, there is no need to repeat it.

• Lines 254 – 256: Is the cumulative incidence expressed as a rate or as a proportion?

• Over all, the discussion section needs a thorough revision.

• It would be better to add a limitation to this study.

6. PLOS authors have the option to publish the peer review history of their article (what does this mean?). If published, this will include your full peer review and any attached files.

Reviewer #1: No

Reviewer #2: No

Reviewer #3: No

---

## [Author Response · Author response to Decision Letter 0]

26 Apr 2023

Thank you for giving this chance to submit revised manuscript. Point-by-point responses for the editor and reviewers comments were provided in a separate file and uploaded to the system as "response to reviewers".

---

## [Decision Letter · Decision Letter 1]

20 Jun 2023

Incidence of switching to second-line antiretroviral therapy and its predictors among children on antiretroviral therapy at general hospitals, Northern Ethiopia: A survival analysis

PONE-D-22-19688R1

Dear Dr. Sibhat,

We’re pleased to inform you that your manuscript has been judged scientifically suitable for publication and will be formally accepted for publication once it meets all outstanding technical requirements.

Kind regards,

Sirinya Teeraananchai

Academic Editor

PLOS ONE

Additional Editor Comments (optional):

Reviewers' comments:

Reviewer's Responses to Questions

**Comments to the Author**

1. If the authors have adequately addressed your comments raised in a previous round of review and you feel that this manuscript is now acceptable for publication, you may indicate that here to bypass the “Comments to the Author” section, enter your conflict of interest statement in the “Confidential to Editor” section, and submit your "Accept" recommendation.

Reviewer #1: All comments have been addressed

2. Is the manuscript technically sound, and do the data support the conclusions?

Reviewer #1: Yes

3. Has the statistical analysis been performed appropriately and rigorously? 

Reviewer #1: Yes

4. Have the authors made all data underlying the findings in their manuscript fully available?

Reviewer #1: Yes

5. Is the manuscript presented in an intelligible fashion and written in standard English?

Reviewer #1: Yes

6. Review Comments to the Author

Reviewer #1: All comments previously has been addressed and incorporated by the author, The manuscripts reads well and its coherent, The metrology is appropriate for the study design, The statistical analysis is also appropriate for the study design and all the assumptions have been considered. The discussion and conclusions are sound, Overall a good paper and contributes to knowledge of Incidence of switching to second-line antiretroviral therapy and its predictors among children on antiretroviral therapy. This is worth publishing

7. PLOS authors have the option to publish the peer review history of their article (what does this mean?). If published, this will include your full peer review and any attached files.

Reviewer #1: No

---

## [Editor Report · Acceptance letter]

30 Aug 2023

PONE-D-22-19688R1 

Incidence of switching to second-line antiretroviral therapy and its predictors among children on antiretroviral therapy at general hospitals, Northern Ethiopia: A survival analysis 

Dear Dr. Sibhat:

I'm pleased to inform you that your manuscript has been deemed suitable for publication in PLOS ONE. Congratulations! Your manuscript is now with our production department. 

Kind regards, 

on behalf of

Dr. Sirinya Teeraananchai 

Academic Editor

PLOS ONE